# Storage Temperature Affects Platelet Activation and Degranulation in Response to Stimuli

**DOI:** 10.3390/ijms26072944

**Published:** 2025-03-24

**Authors:** Ben Winskel-Wood, Denese C. Marks, Lacey Johnson

**Affiliations:** 1Research and Development, Australian Red Cross Lifeblood, Sydney, NSW 2015, Australia; dmarks@redcrossblood.org.au (D.C.M.); ljohnson@redcrossblood.org.au (L.J.); 2Sydney Medical School, The University of Sydney, Sydney, NSW 2050, Australia; 3School of Science, RMIT University, Melbourne, VIC 3000, Australia

**Keywords:** platelets, transfusion, extracellular vesicles, LPS, Histone-H4, cytokines

## Abstract

The refrigeration (cold storage) of platelet components provides several benefits over room-temperature (RT) storage, extending the shelf-life up to 21 days. However, the effect of storage conditions on platelet activation in response to stimulation remains unclear. A paired study was conducted where buffy-coat platelet concentrates were pooled, split, and allocated to RT or cold storage (n = 6 in each group). Platelet samples were taken on days 1, 7, 14, and 21, which were tested without stimulation or following activation with TRAP-6, A23187, lipopolysaccharides, or Histone-H4. Imaging flow cytometry was used to assess the surface characteristics of platelets and extracellular vesicles (EVs). The supernatant concentration of EGF, RANTES, PF4, CD62P, IL-27, CD40L, TNF-α, and OX40L was examined using ELISA. Cold-stored platelets generated a greater proportion of procoagulant platelets and EVs than RT-stored platelets in response to stimulation. The supernatant of cold-stored components contained lower concentrations of soluble factors under basal conditions, suggesting that platelet granules were better retained. Cold-stored platelets released higher concentrations of soluble factors following stimulation with TRAP-6, A23187, or Histone-H4. Only cold-stored platelets responded to lipopolysaccharides. These data demonstrate that cold-stored platelets retain the capacity to respond to stimuli after 21 days of storage, which may facilitate improved functional post-transfusion.

## 1. Introduction

The prompt transfusion of platelet components significantly reduces the risk of patient mortality in acute haemorrhage [1]. Currently, most blood banks store platelet concentrates at room temperature (RT) with constant agitation, limiting the shelf-life to a maximum of 5–7 days, depending on local guidelines due to the risk of bacterial proliferation and transfusion-transmitted bacterial infection [2,3]. The short shelf-life limits the availability of platelet components in geographically isolated medical settings. Further, components stored at RT are affected by the platelet storage lesion, which is a series of deleterious changes affecting platelet activation, metabolism, and soluble factor release [4,5,6,7]. As storage progresses, platelets degranulate, releasing soluble factors. The accumulation of soluble factors in the liquid fraction (supernatant) of the platelet component is linked with a general reduction in haemostatic effectiveness and an increased risk of adverse events post-transfusion [7,8,9,10,11]. As such, there has been renewed interest in evaluating alternative storage methodologies, which allow for the extension of component shelf-life while also better preserving the haemostatic effectiveness of platelet components.

One promising alternative is cold storage, which involves the refrigeration of platelet components at 2–6 °C without agitation. Cold storage is not a new concept; until the 1970s, all platelet components were refrigerated. This practice was discontinued when refrigeration was shown to cause a reduction in the platelet circulation time post-transfusion, reducing the efficacy of cold-stored platelets when transfused prophylactically [12]. Since then, extensive efforts have been made to understand the mechanisms mediating clearance and ameliorate these effects to enable restoration of the in vivo circulation time of cold-stored platelets [13,14,15,16,17,18,19,20]. While this work continues, it has highlighted the advantages that cold-stored platelets may offer in a modern transfusion context, leading to the reconsideration of this product for the treatment of bleeding patients [21,22,23].

Cold storage offers two key advantages over RT platelets. First, refrigeration reduces the rate of bacterial growth [3,24]. Secondly, in vitro data suggest that cold-stored platelets maintain their haemostatic function for up to 21 days [4,5,11]. Together, this may allow for an extension of shelf-life to 14–21 days. The extended shelf-life may enable the supply of platelet components to clinical sites where the maintenance of an inventory of conventionally stored platelets is currently impractical or results in unacceptable wastage due to an infrequent need for transfusions, such as geographically isolated regions.

Cold storage significantly alters the morphology and surface phenotype of platelets [7,11,19,25,26,27]. Shortly after refrigeration, cold-stored platelets undergo extensive morphological changes, which are accompanied by an increase in the abundance of activation markers (PAC-1, P-selectin, and phosphatidylserine) on the surface membrane [25,27,28,29,30]. As P-selectin is stored internally in the alpha (α-) granules, increased surface exposure is often associated with degranulation and the release of other soluble factors. However, despite exhibiting higher surface exposure of P-selectin, the supernatant of cold-stored platelets contains a lower concentration of many soluble factors (RANTES, NAP2, CD62P, PF4, and EGF) under basal conditions compared to RT-stored components [6,7,11,31]. Consequently, cold storage may result in better retention of the granular contents, which could be of functional benefit if released appropriately post-transfusion.

Platelet activation is not a uniform process; different stimuli can generate phenotypically distinct subpopulations and release profiles of soluble factors [32,33,34,35,36]. Depending on the method of activation, platelets can release an extensive collection of soluble factors from α-granules, dense granules, and lysosomes. These factors promote various effects, including haemostasis (PF4), cellular proliferation (EGF), inflammation (RANTES, IL-27), leukocyte activation/differentiation (CD62P, OX40L, CD40L, and TNF-α), and pathogen clearance (PF4) [7,31,37,38]. Additionally, activated platelets can release extracellular vesicles (EVs), which serve as delivery vectors and contribute to haemostasis by providing a catalytic surface for coagulation factors [6,35,39,40].

The effect of refrigeration on the spontaneous release of soluble factors during storage has previously been investigated [6,7,11,19]. However, several of the collection and storage variables known to affect platelet activation, including leukoreduction and additive solutions, differ between these studies. Further, an understanding of how cold storage affects the responsiveness of platelet components to immune and agonist stimulation is required. As such, this study was designed to examine the response of cold-stored platelets to activation signals from both haemostatic and immune stimuli by evaluating changes in the surface phenotype, the generation of EVs, and the release of soluble mediators.

## 2. Results

### 2.1. Platelet Count During Storage

The day 1 baseline platelet count was 1010 ± 102 × 10^9^/L, which decreased during 21 days of storage. At the end of the storage, the platelet count had reduced by approximately 50% (505 ± 92 × 10^9^/L) in the RT and 20% (820 ± 79 × 10^9^/L) in the cold-stored platelets (*p* = 0.0039). As expected from previous literature [7,25,31,41,42], cold storage caused a progressive increase in platelet activation markers, as indicated by an increase in PAC-1 and CD62P-positive platelets (Appendix A).

### 2.2. P-Selectin Exposure on the Platelet Surface Membrane

The release of α-granules is often indirectly assessed by measuring the exposure of P-selectin on the surface membrane of platelets. Compared to RT, cold-stored platelets exhibited a two-fold increase in the abundance of P-selectin on the surface membrane over an extended storage period (Figure 1a). A23187-stimulation induced significantly higher exposure of P-selectin compared to unstimulated controls (Figure 1b). The abundance of P-selectin on the surface membrane was two-fold higher in cold-stored platelets compared to RT following stimulation with A23187 (Figure 1b). Platelets stimulated with TRAP-6 showed a significantly higher abundance of P-selectin compared to unstimulated controls, except on day 21. However, no significant difference was observed in the abundance of inducible CD62P between RT or cold storage (Figure 1c). While Histone-H4 caused P-selectin to increase significantly compared to unstimulated samples, this effect decreased as storage progressed. Further, there was no significant difference observed due to the storage temperature, except for day 7, where the abundance was higher in RT-stored samples (Figure 1d). Cold-stored platelets stimulated with LPS exhibited a significant increase in CD62P on day 7 only compared to both unstimulated and RT samples (Figure 1e); otherwise, P-selectin exposure was comparable to unstimulated samples.

### 2.3. Procoagulant Platelet Subpopulation

The formation of procoagulant platelets is often associated with enhanced thrombin generation and the release of procoagulant phosphatidylserine-positive (PS+) EVs [32,33]. Procoagulant platelets exhibited a spherical or balloon-shaped morphology and stained positive for CD62P, CD42b, and annexin-V (Appendix A). Representative brightfield, fluorescence, and darkfield images are shown for the major platelet subpopulations present in day 1 samples under basal or stimulated conditions (Figure 2a). The proportion of procoagulant platelets was negligible in RT-stored components. However, a gradual increase in the proportion of procoagulant platelets occurred in cold-stored samples over storage, reaching approximately 20% by day 21 (Figure 2b). Platelet activation by A23187 caused approximately 80% of the platelets to transition to a procoagulant phenotype (Figure 2c). Cold-stored platelets maintained this responsiveness throughout 21 days of storage, while RT components exhibited a reduction in the ability to generate procoagulant platelets in response to A23187 from day 14 onwards (Figure 2c). The observed decrease in procoagulant platelets in RT samples was due to a loss of CD42b staining (Appendix A). TRAP-6 stimulation did not induce an increase in the proportion of procoagulant platelets present in RT or cold-stored compared to unstimulated samples (Figure 2d). Histone-H4 induced the formation of a procoagulant phenotype in both RT and cold platelets, and the proportion of procoagulant platelets formed was significantly higher in cold-stored platelets (Figure 2e). Stimulation with LPS increased the percentage of procoagulant platelets in cold-stored, but not RT-stored, components from day 7 onwards (Figure 2f).

### 2.4. Annexin-V-Positive Extracellular Vesicles

In unstimulated platelets, the number of annexin-V+ EVs was higher in cold-stored components compared to RT samples from day 7 onwards (Figure 3a). Stimulation with A23187 induced the release of significantly more annexin-V+ EVs compared to unstimulated controls (Figure 3b). In general, the number of annexin-V+ EVs was two-fold higher in cold-stored platelets activated with A23187 compared to RT (Figure 3b). Activation with TRAP-6 caused an increase in the number of annexin-V+ EVs compared to unstimulated samples. However, the difference in the concentration of annexin-V+ EVs in RT and cold-stored samples was not statistically significant (Figure 3c). Histone-H4 stimulation initiated the release of annexin-V+ EVs from day 1 and cold-stored platelets (Figure 3d), although a decrease occurred as storage progressed, reaching numbers similar to day 1, by day 14 in cold-stored samples. In comparison the number of EVs released from RT-stored platelets activated with Histone-H4 was minimal (Figure 3d). Incubation with LPS caused the release of a greater number of annexin-V+ EVs from cold-stored platelets compared to both unstimulated cold-stored and LPS-stimulated RT-stored platelets (Figure 3e). As such, the overall trend was for cold-stored platelets to release higher numbers of annexin-V+ EVs over storage and in response to stimuli.

### 2.5. Soluble Factor Concentration in Platelet Components

The supernatant of cold-stored platelets contained between two- and four-fold lower concentrations of EGF, RANTES, PF4, CD62P, and IL-27 compared to components stored at RT (Table 1). No differences in the concentration of CD40L, TNF-α, or OX40L were observed between RT and cold-stored platelets over the 21-day storage period (Table 1).

### 2.6. Correlation Between Platelet Count and Soluble Factor Release

There was a significant correlation between the reduction in platelet count and the spontaneous release of EGF, RANTES, PF4, CD62P, IL-27, CD40L, and TNF-α but not OX40L during storage (Figure 4).

### 2.7. The Concentration of EGF, RANTES, PF4, and CD62P

All samples activated with A23187, TRAP-6, and Histone-H4 released measurable concentrations of EGF, RANTES, PF4, and CD62P (Figure 5a–d). Stimulation with LPS led to the specific release of PF4 from cold-stored platelets (Figure 5c). Platelets activated with A23187 released EGF at similar concentrations regardless of the storage time or temperature (Figure 5a). Cold-stored platelets released higher concentrations of EGF following stimulation with TRAP 6 on days 7–14 but not on day 21 compared to RT (Figure 5a). RT and cold-stored platelets activated with Histone-H4 released similar concentrations of EGF, which were significantly lower than day 1 (Figure 5a). Cold-stored platelets better maintained the capacity to release RANTES following activation with A23187, TRAP-6, and Histone-H4 compared to RT components (Figure 5b). In general, the concentration of RANTES released by RT but not cold-stored platelets declined significantly by day 7 (A23187) or day 14 (TRAP-6 and Histone-H4) compared to day 1 (Figure 5b). Overall, cold-stored platelets released significantly higher concentrations of PF4 in response to A23187, Histone-H4, and LPS compared to RT samples (Figure 5c). The exception was following TRAP-6 stimulation, where PF4 release was equivalent between temperatures. The storage temperature did not impact the concentration of CD62P released from platelets following activation with A23187 or TRAP-6. However, Histone-H4 stimulated cold-stored platelets released significantly higher concentrations of CD62P compared to RT from day 14 onwards (Figure 5d).

### 2.8. The Concentration of IL-27, CD40L, TNF-α, and OX40L

Overall, the release of IL-27, CD40L, TNF-α, and OX40L was comparatively low in response to the applied stimuli (Figure 6). LPS did not induce the significant release of any of these mediators, regardless of the storage temperature or time. An increase in IL-27 was observed in RT-stored components activated with LPS, but the change was not statistically significant compared to cold-stored platelets or day 1 (Figure 6a). There was no significant difference in the concentration of IL-27 in the supernatant of RT or cold-stored platelets following activation with A23187 or TRAP-6 (Figure 6a). However, cold-stored platelets were observed to release higher concentrations of IL-27 compared to RT on day 21 of storage in response to Histone-H4. Cold-stored platelets released higher concentrations of CD40L after activation with A23187, TRAP-6, and Histone H4 compared to RT-stored platelets and day 1 samples until the end of storage (Figure 6b). Detectable concentrations of TNF-α were released from A23187 and Histone-H4-activated platelets, but there was no statistical difference between the storage time or temperature (Figure 6c). All stimuli induced OX40L release, but no significant differences were observed between samples (Figure 6d).

## 3. Discussion

The data presented in this study demonstrate the impact of the ex vivo storage temperature on platelet function and soluble factor release. Platelets are capable of dynamically responding to a broad range of agonists, which, depending on the stimuli, can result in the formation of phenotypically distinct subpopulations and the release of an extensive number of soluble factors from internally stored granules [6,43]. In line with previous work [6,7,11,31], cold-stored platelets exhibited a more activated phenotype, accompanied by a reduced accumulation of soluble factors under basal conditions, compared to RT components. Further, our findings suggest that cold-stored platelets are better able to release soluble factors and annexin-V+ EVs in response to both haemostatic (TRAP-6 and A23187) and immune (LPS and Histone-H4) stimuli compared to RT components. These findings suggest that the granular content of cold-stored platelets is better retained during extended storage, potentially explaining why a higher concentration could be released into the supernatant following activation.

In this study, we observed significant variations in the composition of the releasate depending on how the platelets were stored and activated. Platelets, once considered simple cells, are now recognised as being highly dynamic, possessing intricate signalling pathways that allow them to release the contents of several granular compartments (α-granules, dense granules, and lysosomes) in response to a wide variety of haemostatic and immune stimuli [43,44]. These responses can lead to diversity in the composition of soluble factors released [6,32,35,36,44]. While the exact mechanisms mediating the variations in factors released are still under investigation, previous work has highlighted that the function of α-granules may be crucial. The α-granules are believed to contain over 300 separate proteins, many of which can be selectively packaged with cargo by the parent megakaryocyte prior to platelet formation [43]. Unlike lysosomes or dense granules, α-granules can fuse with one another before release. This fusion, known as compound exocytosis [44], allows for the release of either individual granules or larger fused structures. The mode of α-granule release appears to depend on both the type and concentration of the agonist, with stronger stimuli favouring fusion and compound exocytosis [44]. This capability for the selective release of soluble factors in response to different stimuli provides a potential explanation for the variations observed in this study and others.

In line with a previous publication [11], a clear relationship was observed between the platelet count and the baseline concentration of soluble factors in the supernatant. On day 21 of storage, RT-stored components had a higher baseline concentration of soluble factors, which coincided with a 50% decrease in the platelet count. Notably, the accumulation of pro-inflammatory soluble factors (RANTES and CD40L) in the supernatant of RT-stored components is associated with a reduction in haemostatic function in vitro and an increased risk of adverse events post-transfusion [8,11,38,45]. Consequently, the transfusion of cold-stored platelets, which exhibit a lower accumulation of pro-inflammatory soluble factors, may carry a lower risk of adverse events, although additional clinical data are required to substantiate this. Traditionally, platelet degranulation and soluble factor release are caused by agonist-induced activation [32]. However, the accumulation of soluble factors during extended storage at RT is believed to be driven by metabolic disruption and the upregulation of apoptotic pathways, leading to a reduction in membrane integrity and the potential release of the cytoplasmic contents [5,46,47]. In contrast, cold-stored platelets retain membrane integrity, likely due to better preservation of metabolic health, including glucose stores and other metabolites [4,42,48]. These findings underscore the role of storage conditions in influencing platelet viability and the release of soluble factors over time.

As expected, platelets were most responsive to stimuli on day 1 of storage [4,31,42,49], which gradually decreased as storage progressed. Notably, cold-stored platelets retained the ability to respond to agonist stimulation until the end of storage (day 21), albeit at a reduced capacity. Platelet activation and the localised release of soluble factors at the site of injury play a vital role in promoting cellular proliferation (EGF), immune activation (IL-27, CD40L, and CD62P), and inflammation (IL-27 and RANTES), which support tissue repair and pathogen clearance [50]. Given their enhanced ability to retain and release these factors, it is theorised that cold-stored platelets may be more effective in facilitating these processes upon transfusion than RT-stored components. In comparison, RT-stored platelets exhibited a more rapid reduction in agonist responsiveness and soluble factor release over time. Platelet activation and the release of the granular contents are energy-intensive processes that depend on the rapid generation of ATP in the mitochondria, fuelled by extracellular glucose [51]. While not examined in this study, previous work shows that cold storage reduces cellular respiration, preserving metabolic parameters and glucose stores, which are often depleted in RT-stored components [4,46,48]. This preservation of metabolic integrity may explain why cold-stored platelets maintain a greater capacity to release soluble factors in response to stimuli compared to their RT-stored counterparts.

Cold-stored components contain a higher proportion of PS+ procoagulant platelets and PS+ EVs at baseline compared to RT-stored samples. Upon stimulation with A23187, histone-H4, or LPS, cold-stored platelets generate even more PS+ procoagulant platelets and EVs than RT-stored components. The presence of PS on the platelet surface facilitates the coagulation cascade by catalysing thrombin generation, which has been associated with reduced clot formation time (R-time, thromboelastography; time to clot, and turbidimetric clotting analysis) in vitro [6,39,40]. The potential clinical impact of increased procoagulant activity post-transfusion remains poorly understood. However, in vivo, procoagulant platelets and EVs have been linked to both beneficial haemostatic effects and thrombotic complications [33,40]. Considering that cold-stored platelets are primarily intended for the management of acute bleeding, such as in traumatic injury or cardiac surgery, further investigation into their procoagulant function is essential. Ongoing clinical trials, including CoVeRTS-HM (NCT05820126), PLTS-1 (NCT06147531), 4CPLT (NCT02495), and CHIPS (NCT04834414), which aim to evaluate the efficacy and safety of cold-stored platelets in treating active bleeding may provide further insight. Although completed trials demonstrate safety [52,53].

In addition to promoting procoagulant function, platelet EVs can act as delivery vectors for a broad range of soluble factors (IL-27, EGF, RANTES, CD40L, and CD62P), mitochondria, mRNA, and surface-bound receptors or lipids from the parent cell [8,36,39,41]. The contents of EVs are highly variable, dependent on the activating factor, and can differ significantly from the soluble factors present in the parent cell [35]. Depending on their contents, EVs are capable of promoting cellular growth (EGF), immune activation (CD62P, CD40L, and IL-27), and inflammation (RANTES and IL-27) [9,37,39]. In a transfusion context, high concentrations of EVs containing pro-inflammatory factors (mitochondrial mRNA) have been directly linked with an increased risk of adverse events post-transfusion [9]. Although the contents of EVs were not examined in this study, further investigation is warranted to identify the extent to which storage time, temperature, and stimuli influence the contents and function of EVs.

We have shown that cold storage affects the platelet responses to activation, and this likely occurs through altered signal transduction. Different stimuli were used to explore distinct facets of storage-induced platelet activation. A23187 and TRAP-6 are commonly used to investigate the haemostatic function of platelets [30,32,33]. In contrast, Histone-H4 and LPS, which are less studied in a transfusion context, are immunological stimuli, which also facilitate platelet activation. Histone-H4 is released from the nuclei of damaged cells and is associated with the formation of procoagulant platelets and thrombocytopenia in patients suffering from traumatic injury [32], whereas LPS from Gram-negative bacteria can cause platelet activation and soluble factor release through the immune receptor TLR4 [36]. Of particular interest, activation with LPS was exclusively seen in cold-stored platelets. A small but significant increase in procoagulant platelet formation, P-selectin exposure, and the release of PS+ EVs was observed, but degranulation was limited to PF4 release. While the effect of LPS on stored platelets remains largely unstudied, previous research from our laboratory indicates that cold storage enhances platelet sensitivity to LPS-expressing *Escherichia coli*, leading to increased aggregation [25]. Additionally, recent studies have shown that even short periods of refrigeration (1 h) heighten platelet sensitivity to aggregation induced by haemostatic agonists such as ADP, collagen, and TRAP-6 [30]. This heightened sensitivity was associated with increased phosphorylation of signalling proteins (ERK-1/ERK-2, p38-MAPK, and Akt) and reduced responsiveness to inhibitory signalling molecules such as prostaglandin E2 and DEA/NO [30]. Importantly, LPS-induced platelet activation via TLR4 depends on signalling through p38-MAPK and Akt [30,36]. Furthermore, cold storage elevates the abundance of P-selectin and fibrinogen on the platelet surface membrane [7,26,29], which can bind to LPS [54,55]. Together, these findings suggest that cold-induced alterations may enhance sensitivity to non-haemostatic agonists such as LPS.

P-selectin is an α-granule protein, which is mobilised to the surface membrane or released directly into the supernatant following activation and granular exocytosis [44,56]. As such, P-selectin exposure on the surface of platelets is often used as a measure of platelet activation and degranulation [6,51]. However, P-selectin can be cleaved from the surface membrane by proteolysis, through matrix metalloproteinases, or by transfer to EVs [40,56]. In this study, RT-stored platelets exhibited a lower surface abundance of P-selectin, which, when interpreted in isolation, suggests that they are less activated. However, the supernatant of RT platelets contained 2–4-fold higher concentrations of soluble factors compared to cold-stored samples, implying greater activation and degranulation. As such, the discrepancy between the abundance of P-selectin on the surface and in the supernatant of RT-stored platelets is likely due to increased removal from the surface membrane during storage. Consequently, measuring P-selectin membrane exposure alone may not represent the best protein to indicate degranulation.

While this study presents several novel and interesting findings, there are also limitations. Although a relatively small sample size (six replicates) was tested, the biological variability was minimised by conducting a paired study. By pooling two platelet components (each containing buffy coats of four whole-blood donations), each replicate contained the platelets from eight individual donors. Another consideration is that differences in collection methods and storage solutions (plasma vs. PAS) are known to significantly impact platelet metabolic parameters and the degree of activation during storage [33,41,46,57,58]. As such, these results may not be representative of platelet responses stored under different conditions. Additionally, as the focus of this study was to examine how storage affects the sensitivity of platelets to activation and degranulation rather than functional changes, our conclusions are based on a limited number of in vitro parameters. Finally, in vitro experiments are unlikely to fully reflect the dynamics of platelet activation post-transfusion.

Further research is needed to assess the in vivo effects of cold-stored platelets and determine the optimal shelf-life. These data support and extend that of others, showing that cold-stored platelets retain responsiveness to clinically relevant stimuli, which may be beneficial post-transfusion.

## 4. Materials and Methods

### 4.1. Study Design

Ethics approval was obtained from the Australian Red Cross Lifeblood Ethics Committee prior to commencement of this study (Johnson 10052019). All donations utilised in this study were obtained from voluntary, non-remunerated donors. Pooled platelet concentrates were manufactured by combining the buffy coats from four whole blood donations, which were resuspended in 30% plasma/70% additive solution (PAS-E, SSP+; Macopharma, Tourcoing, France). The platelet components were leukoreduced using AutoStop BC filters and stored in ELX 1300 mL PVC bags (Haemonetics, Boston, MA, USA). Each replicate consisted of two pooled platelet concentrates, which were combined and split into two components of equal weight on day 1 post-collection (Appendix A). One of each pair was randomly assigned to storage at RT (20–24 °C) with agitation (Helmer, Noblesville, IN, USA) or refrigerated storage (cold storage, 2–6 °C) without agitation. Approximately 10–15 mL was removed from the combined component on day 1 to establish baseline parameters and from RT or cold-stored components on days 7, 14, and 21 post-collection (Appendix A). Platelet counts were obtained using an automated haematology analyser (CellDYN Emerald 22, Abbot Core Laboratory, Abbott Park, IL, USA).

### 4.2. Platelet Stimulation

All agonists were purchased from Merck Life Sciences, Darmstadt, Germany. Platelet samples were diluted in annexin-V binding buffer (Biolegend, San Diego, CA, USA; filtered using a 0.1 μm PVDF membrane, Merk-Millipore prior to use) to ensure sufficient extracellular calcium was available prior to activation. Platelets were diluted to 300 × 10^6^/mL and left unstimulated or activated with TRAP-6 (10 µM), A23187 (10 µM), and lipopolysaccharide (LPS, 20 µg/mL) from *Escherichia Coli* (*E. coli*, O111:B4; Merck Life Sciences, Darmstadt, Germany). For activation with Histone-H4 (30 µg/mL; Merck Life Sciences), platelets were diluted to 100 × 10^6^/mL to prevent serum albumin-mediated inhibition of platelet activation [59]. Platelet samples were mixed and incubated for either 15 (A23187 and Histone-H4) or 30 min (TRAP-6 and LPS) at 37 °C prior to analysis [32,33,36]. The same samples were used for both imaging flow cytometry experiments and the preparation of supernatants for soluble factor analysis.

### 4.3. Imaging Flow Cytometry

To characterise the phenotype, diluted platelet samples were stained with a panel consisting of PAC-1-FITC (PAC-1; BD Bioscience, Franklin Lakes, NJ, USA), CD62P-PE (AC1.2; BD Bioscience), CD42b-PE Dazzle 594 (HIP-1; Biolegend), and Annexin-V-APC (Biolegend). Samples were incubated for 20 min in the dark before dilution with 150 µL annexin-V binding buffer (Biolegend). Samples were analysed using an imaging flow cytometer (Imagestream^X^ Mk II; Cytek, Fremont, CA, USA) equipped with one charge-coupled device (CCD) camera, two excitation lasers (488 nm: 100 mW, 642 nm: 150 mW), and a side scatter laser (785 nm: 2 mW). Data were acquired based on brightfield (area and aspect ratio) and darkfield (scatter) parameters with 7500 platelet events collected at a low flow rate at 60× magnification.

Imaging flow cytometry data were analysed using IDEAS v6.2 software (Cytek, Fremont, CA, USA). The data collection gates and post-collection masks used for analysis were set based on a combination of unstained and fluorescence minus one (FMO) controls. Platelet and EV gates were generated with reference to known size beads (200 nm and 1 μm, Invitrogen) and Verity Shells (184 nm and 389 nm; Exometry, Amsterdam, The Netherlands). Compensation matrices were created from single-stained platelets activated with TRAP-6 (10 μM; PAC-1-FITC, CD62P-PE; CD42b-PE Dazzle 594) or A23187 (10 μM; annexin-V-APC). TRAP-6- and A23187-stimulated platelets were used to generate compensation controls, as activation by these factors resulted in the highest abundance of the listed markers on the platelet surface membrane. Post-collection data analysis was facilitated by the creation of masks to identify platelet subpopulations and distinguish cellular events and EVs from debris. EVs were differentiated from LPS micelles via brightfield (area and aspect ratio), darkfield (scatter), and fluorescence (CD42b-PE Dazzle 594) parameters (Appendix A). Fluorescence and morphology data were used to identify the procoagulant platelet subpopulation (PAC-1-/CD62P+/CD42b+/Annexin-V+) in line with the current literature [32,33,34]. Platelet EVs were classified as particles between 0.1 and 0.8 μm in size, which were dual-positive for CD42b-PE Dazzle 594 and annexin-V-APC.

### 4.4. Soluble Factor Analysis

Platelet supernatant was collected at each time point using double centrifugation, as previously described [4]. Briefly, unstimulated or stimulated platelet samples were centrifuged sequentially at 1600× *g* for 20 min, followed by 12,000× *g* for 5 min. The resulting supernatant was frozen and stored at −80 °C for later analysis. The concentration of soluble factors in the supernatant was investigated using commercially available enzyme-linked immunosorbent assay (ELISA): EGF, RANTES, PF4, CD62P, IL-27, CD40L, TNF-α, and OX40L (R&D Systems, Minneapolis, MN, USA). The limit of detection was listed as 4 pg/mL (EGF), 16 pg/mL (RANTES, PF4, and CD40L), 125 pg/mL (CD62P), 156 pg/mL (IL-27), 31 pg/mL (TNF-α), and 47 pg/mL (OX40L). Each sample was tested in duplicate or triplicate and measured against a standard curve, as per the manufacturer’s instructions. The resulting soluble factor concentration was normalised to the platelet dilution of the stimulated samples (i.e., multiplied by a factor of three for Histone H4). The change (Δ) in soluble factor concentration was calculated by subtracting the baseline concentration of soluble factor obtained from unstimulated supernatant from the corresponding stimulated samples.

### 4.5. Statistical Analysis

The data are presented as the mean ± the standard deviation (SD). Statistical analysis was conducted using GraphPad Prism 9.4.1 (GraphPad Software, Inc., La Jolla, CA, USA). A two-way repeated-measures analysis of variance (ANOVA) was used to compare the effects of temperature (RT and cold), storage time, and stimulation on platelet samples over the storage period. Post hoc Bonferroni multiple comparisons were performed to determine the differences between each treatment at each time point. The Pearson correlation coefficient was used to determine the correlation between platelet count and the concentration of soluble mediators. In all statistical tests, a *p*-value < 0.05 was considered to be significant.

## Figures and Tables

**Figure 1 ijms-26-02944-f001:**
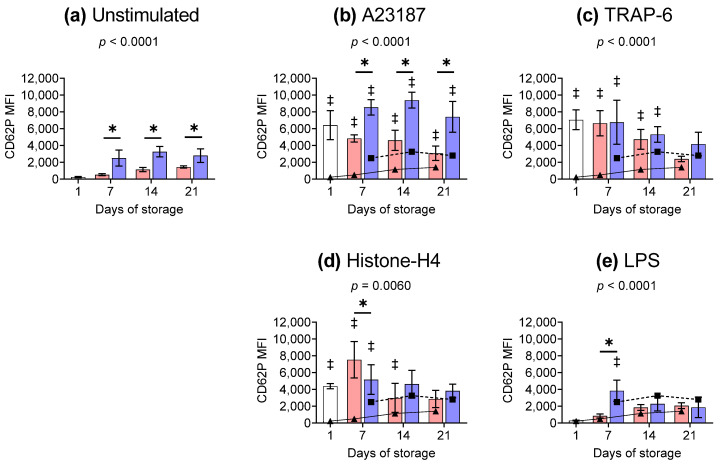
The exposure of P-selectin on the platelet surface membrane is affected by both storage temperature and stimuli. Platelets were sampled on day 1 (
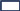
) or following room-temperature (
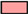
) or cold storage (
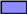
) on days 7, 14, and 21 post-collection. The median fluorescence intensity (MFI) of CD62P-PE is shown for (**a**) unstimulated platelets or samples activated with (**b**) A23187, (**c**) TRAP-6, (**d**) Histone-H4, or (**e**) LPS. Data represent the mean ± standard deviation (error bars, n = 6). The CD62P MFI from (**a**) unstimulated samples at RT (
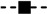
) or 4 °C (
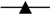
) has been overlaid on the corresponding (**b**–**e**) stimulated samples for comparison. Platelets were stained with PAC-1-FITC, CD62P-PE, CD42b-PE-Dazzle 594, and annexin-V-APC and analysed using imaging flow cytometry at 60× magnification. Platelet events were gated based on size (area and aspect ratio) and scatter (darkfield). Significance was determined using two-way ANOVA comparing the effects of temperature (RT vs. 4 °C) and stimulation on platelet samples over time, with the interaction *p*-value presented. * = *p* < 0.05 compared to room temperature at the same time point. ‡ = *p* < 0.05 compared to unstimulated platelets at the same time point.

**Figure 2 ijms-26-02944-f002:**
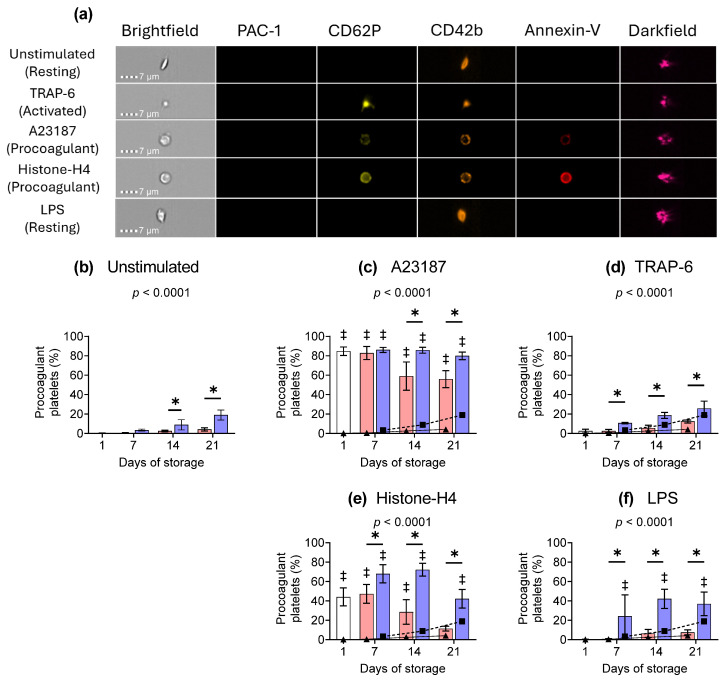
Cold storage promotes a procoagulant phenotype during storage and after exposure to stimuli. Platelets were sampled on day 1 (
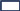
) or following room-temperature (
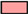
) or cold storage (
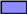
) on days 7, 14, and 21 post-collection. (**a**) Representative brightfield, fluorescence, and darkfield images are shown for the major platelet subpopulations present in day 1 samples. The percentage of procoagulant platelets present in (**b**) unstimulated or following activation with (**c**) A23187 (10 µM), (**d**) TRAP-6 (10 µM), (**e**) Histone-H4 (30 µg/mL), or (**f**) LPS (20 µg/mL) is shown over storage. The percentage of procoagulant platelets present in (**a**) unstimulated samples at RT (
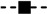
) or 4 °C (
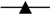
) are overlaid on the corresponding (**b**–**e**) stimulated samples for comparison. Platelets were stained with PAC-1-FITC, CD62P-PE, CD42b-PE-Dazzle 594, and annexin-V-APC and analysed using imaging flow cytometry at 60× magnification. Platelet events were gated based on size (area and aspect ratio) and scatter (darkfield). Data represent the mean ± standard deviation (error bars, n = 6). Significance was determined using two-way ANOVA comparing the effects of temperature (RT vs. 4 °C) and stimulation on platelet samples over time, with the interaction *p*-value presented. * = *p* < 0.05 compared to room temperature at the same time point. ‡ = *p* < 0.05 compared to unstimulated platelets at the same time point.

**Figure 3 ijms-26-02944-f003:**
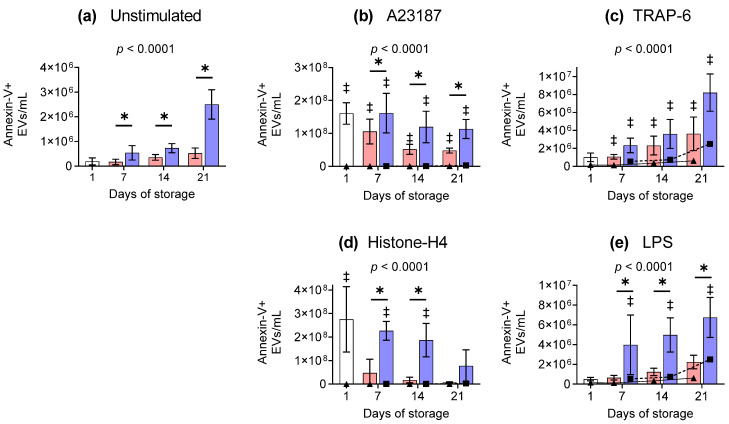
Cold-stored platelets release higher concentrations of annexin-V-positive extracellular vesicles in response to stimuli. Platelets were sampled on day 1 (
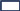
) or following room-temperature (
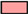
) or cold storage (
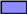
) on days 7, 14, and 21 post-collection. The concentration of extracellular vesicles in (**a**) unstimulated platelets or following activation with (**b**) A23187 (10 µM), (**c**) TRAP 6 (10 µM), (d) Histone-H4 (30 µg/mL), or (**e**) LPS (20 µg/mL). Data represent the mean ± standard deviation (error bars, n = 6). The number of EVs from a) unstimulated samples at RT (
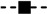
) or 4 °C (
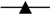
) is overlaid on the corresponding (**b**–**e**) stimulated samples for comparison. Platelet samples were stained with PAC 1 FITC, CD62P-PE, CD42b-PE-Dazzle 594, and annexin-V-APC and analysed using imaging flow cytometry at 60× magnification. Extracellular vesicle events were gated based on size (area and aspect ratio), scatter (darkfield), and dual positivity for both GPIbα (CD42b) and phosphatidylserine (PS, annexin-V). Significance was determined using two-way ANOVA comparing the effects of temperature (RT vs. 4 °C) and stimulation on platelet samples over time, with the interaction *p*-value presented. * = *p* < 0.05 compared to room temperature at the same time point. ‡ = *p* < 0.05 compared to unstimulated platelets at the same time point.

**Figure 4 ijms-26-02944-f004:**
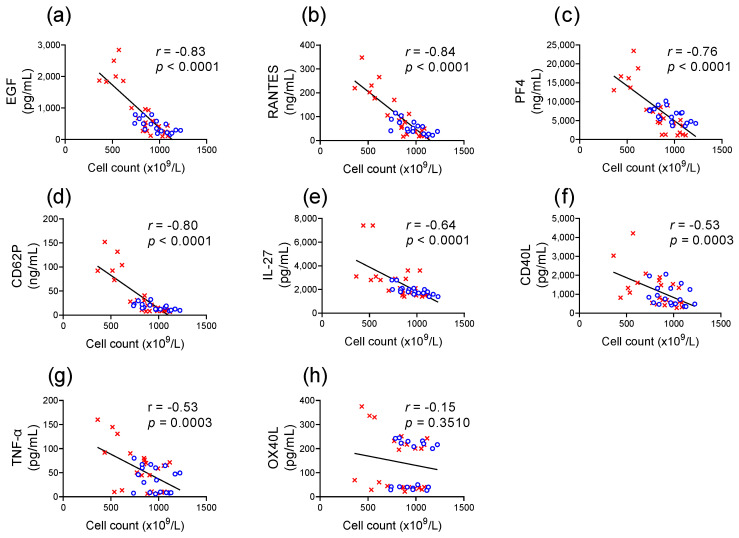
The reduction in platelet count during storage correlates with an increase in the concentration of soluble factors in the supernatant. Individual data points for the platelet count of room-temperature (
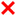
) and cold-stored (
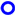
) platelet components are plotted against the concentration of soluble factors in the supernatant for (**a**) EGF, (**b**) RANTES, (**c**) PF4, (**d**) CD62P, (**e**) IL-27, (**f**) CD40L, (**g**) TNF-α, and (**h**) OX40L. Pearson test was used to determine the correlation (*r*-value), and linear regression was used to apply a line of best fit.

**Figure 5 ijms-26-02944-f005:**
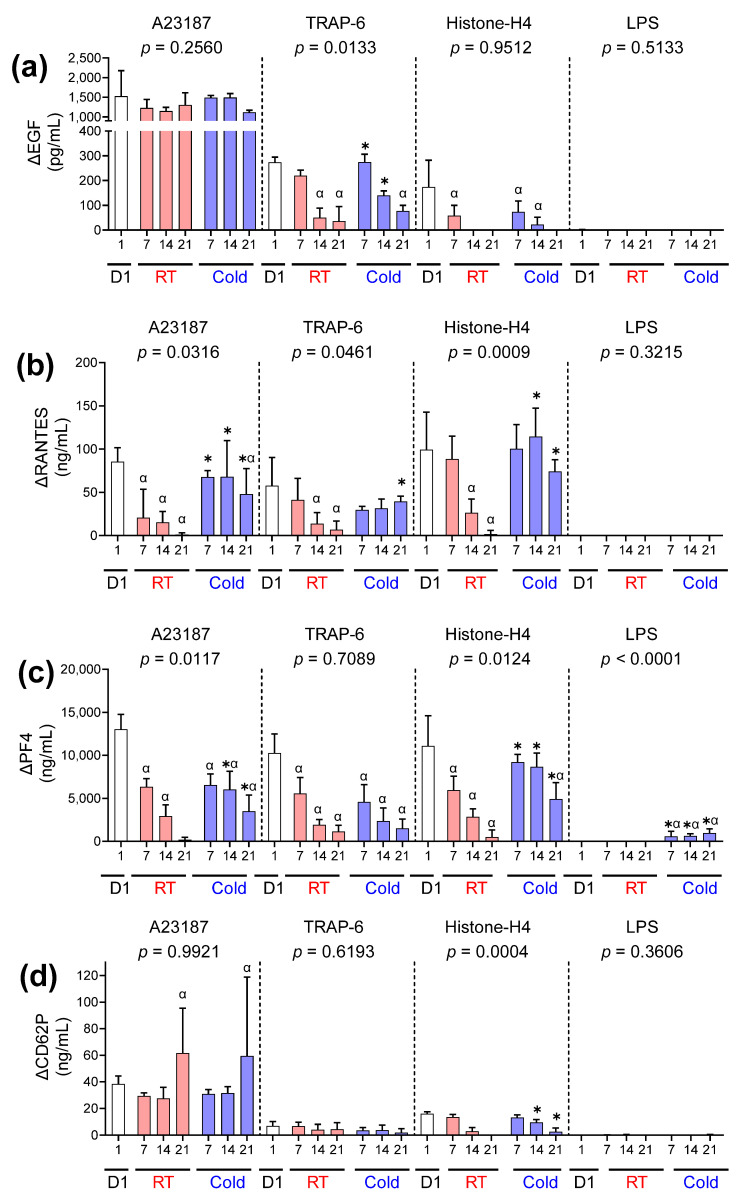
Cold storage alters the release of EGF, RANTES, PF4, and CD62P in response to haemostatic and immune stimuli. Platelets were sampled on day 1 (D1; 
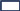
) or following room temperature (
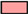
) or cold storage (
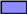
) on days 7, 14, and 21 post-collection (x-axis). The concentration of (**a**) EGF, (**b**) RANTES, (**c**) PF4, and (**d**) CD62P was measured in the supernatant of unstimulated platelets or following activation with A23187 (10 µM), TRAP 6 (10 µM), Histone-H4 (30 µg/mL), or LPS (20 µg/mL) using ELISA. The change (Δ) in soluble factor concentration was calculated by subtracting the concentration of soluble factors obtained from unstimulated supernatant from the corresponding stimulated samples. Data represent the mean ± standard deviation (error bars, n = 6). Significance was determined using two-way ANOVA comparing the effects of temperature (RT vs. 4 °C) and stimulation on platelet samples over time, with the interaction *p* value presented. * = *p* < 0.05 compared to room temperature at the same time point. α = *p* < 0.05 compared to day 1 platelets.

**Figure 6 ijms-26-02944-f006:**
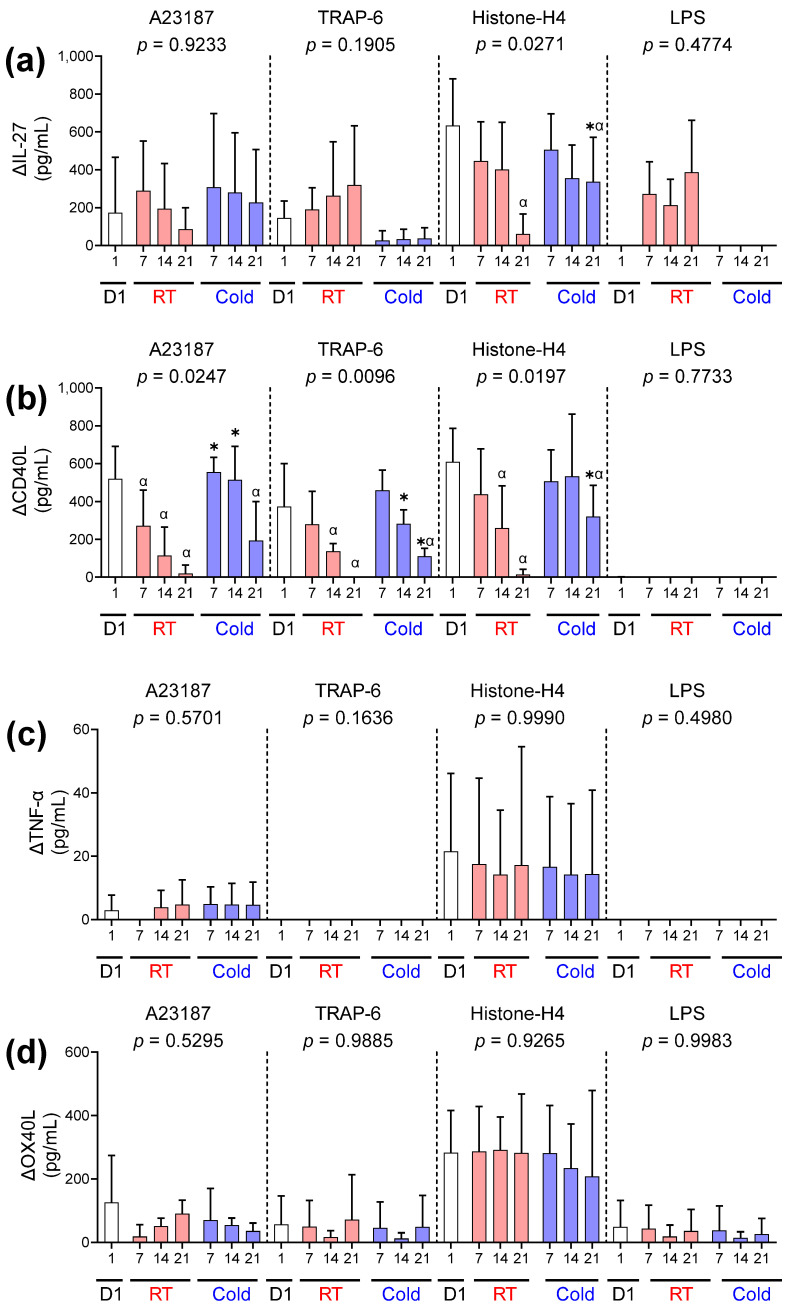
Cold storage alters the release of IL-27 and CD40L but not TNF-α or OX40L in response to haemostatic and immune stimuli. Platelets were sampled on day 1 (D1; 
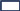
) or following room temperature (
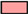
) or cold storage (
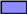
) on days 7, 14, and 21 post-collection. The concentration of (**a**) IL-27, (**b**) CD40L, (**c**) TNF-α, and (**d**) OX40L was measured in the supernatant of unstimulated platelets or following activation with A23187 (10 µM), TRAP-6 (10 µM), Histone-H4 (30 µg/mL), or LPS (20 µg/mL) using ELISA. The change (Δ) in soluble factor concentration was calculated by subtracting the concentration of soluble factors obtained from unstimulated supernatant from the corresponding stimulated samples. Data represent the mean ± standard deviation (error bars, n = 6). Significance was determined using two-way ANOVA comparing the effects of temperature (RT vs. 4 °C) and stimulation on platelet samples over time, with the interaction *p*-value presented. * = *p* < 0.05 compared to room temperature at the same time point. α = *p* < 0.05 compared to day 1 platelets.

**Table 1 ijms-26-02944-t001:** The supernatant of unstimulated cold-stored platelets contains lower concentrations of soluble factors during storage.

		Day of Storage				
Soluble Factor	Storage Method	Day 1 (Baseline)	Day 7	Day 14	Day 21	*p*-Value
EGF (pg/mL)	RT	172 ± 71	347 ± 95	783 ± 203	2150 ± 421	0.0133
	Cold		248 ± 75	397 ± 146 *	668 ± 139 *	
RANTES (ng/mL)	RT	19 ± 3	50 ± 18	104 ± 38	241 ± 60	<0.0001
	Cold		39 ± 22	61 ± 27 *	68 ± 31 *	
PF4 (ng/mL)	RT	1324 ± 276	4318 ± 549	7487 ± 1275	16,990 ± 3788	<0.0001
	Cold		4202 ± 488	6423 ± 691	8691 ± 982 *	
CD62P (ng/mL)	RT	8 ± 1	12 ± 2	29 ± 8	108 ± 29	<0.0001
	Cold		11 ± 1	15 ± 3 *	25 ± 5 *	
IL-27 (pg/mL)	RT	2167 ± 1111	1733 ± 288	2167 ± 575	4433 ± 2302	0.0341
	Cold		1600 ± 179	1933 ± 186	2200 ± 473 *	
CD40L (pg/mL)	RT	695 ± 557	898 ± 572	1123 ± 684	2018 ± 1329	0.5250
	Cold		697 ± 445	857 ± 474	1185 ± 656	
TNF-α (pg/mL)	RT	45 ± 26	45 ± 30	50 ± 35	92 ± 66	0.4530
	Cold		35 ± 22	30 ± 27	46 ± 31	
OX40L (pg/mL)	RT	129 ± 100	119 ± 93	135 ± 110	200 ± 163	0.8727
	Cold		121 ± 96	133 ± 101	138 ± 111	

Values are shown as mean ± SD; n = 6 in each group; *p*-value (interaction) calculated using two-way ANOVA comparing the effects of temperature (RT vs. 4 °C) and stimulation on platelet samples over time. * *p* < 0.05 compared to RT at the same time point. RT, room temperature; cold, cold-stored.

## Data Availability

The raw data supporting the conclusions of this article will be made available by the authors upon request.

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
