# Peer review of "Storage Temperature Affects Platelet Activation and Degranulation in Response to Stimuli"

_ijms, 2025, doi:10.3390/ijms26072944_

Round 1
Reviewer 1 Report
Comments and Suggestions for Authors
Comments to the Author - Wood et al., have carried out a straightforward in vitro study to evaluate how prolonged cold storage affects the responsiveness of platelets to agonist stimulation. Consequently, soluble factor release has been given specific attention. Cold storage is an alternative storage method for platelets that was re-examined in the early 2000s after being abandoned in the 1970s. In recent years, interest has increased and research into cold storage has accelerated in scope. The presented study contains relatively few replicates and suffers from uncontrolled variables but is otherwise well conducted and contains some new information that is likely to be of interest to readers. However, although the study presents some new and interesting findings, there are some concerns regarding the treatment of prior research. The manuscript asserts that many of the recent published data on cold storage are new information, yet similar results were established in previous research. The omission or downplaying of these prior studies gives a misleading impression of originality.
Specific comments
- While some relevant studies are mentioned, they are not appropriately integrated into the discussion. Instead, they appear to be underestimated in a way that downplays their significance to promote recent work. A robust scientific contribution requires acknowledging and building upon previous knowledge rather than presenting well-established findings as recently published or new conclusions. Some of the most foundational works since the renewed interest in cold storage during the early 2000s, appear to have been either omitted, misinterpreted or taken out of context. The manuscript does not acknowledge how key earlier studies laid the foundation for today’s research on cold storage. This permeates both the introduction and the discussion.
Introduction
Page 2; line 43: The authors describe cold storage as a concept that was abandoned in the 1970s (due to shortened circulation time) and recently revived with a stream of newly published "findings". Cold-stored platelets were abandoned not only because of their shorter circulation time in vivo but also because of their short storage time (72hr). Interest in cold-stored platelets was revived in the early 2000s and the following decade, This research period highlighted the potential of cold-stored platelets, paving the way for further studies into extending platelets shelf life under refrigerated conditions. Italiano et.al, The Hoffmeister group, Sorensen and Wandall made significant contributions to the newfound interest, understanding and potential use of cold-stored platelets during the 2000s. It was also shown for the first time that it is possible for sedimented platelets to withstand storage in the cold for up to 21 days (Sandgren Transfusion/Vox Sang 2006/2007). Previous paradigms were under agitation with a maximum cold storage time of 72 hours. Pioneering studies of the phenomenon of cold induced activation were carried out even earlier than the examples by Hartwig and Winokur in the 1990th
Please either rewrite the text so that it gives the reader a fairer picture of the field or add relevant references to the text, throughout where the citations are not of a novel nature
Page 2 line 46; “in modern transfusion context, advantages”
It´s important to note while cold storage significantly reduces bacterial growth, it does not eliminate all bacteria
Page 2 line 51“reducing logistic burden”
Refrigerating a portion of the production may pose logistical challenges.
- Reduced Availability for Patients – Platelets are already in high demand, and reducing the number of universal transfusable units means fewer patients can receive life-saving treatments.
- Mismatch Between Supply and Demand – Platelet demand fluctuates daily, and hospitals rely on a steady supply of fresh platelets. By cooling for example 30%, you risk having an excess of refrigerated platelets that fewer patients can use, while fresh platelets become scarcer.
- Increased Waste – If the cooled platelets are not compatible with enough patients' needs (e.g., because of stricter transfusion criteria), a significant portion might expire before use, leading to unnecessary waste.
- Complicated Inventory Management – Hospitals and blood banks would have to track two separate platelet storage systems: one with a shorter shelf life and wider usability, and one with a longer shelf life but limited application. This adds complexity and increases the risk of mismanagement.
- Increased Costs – Managing two platelet storage systems requires additional training, monitoring, and possibly new refrigeration equipment. These costs may outweigh the benefits of extending shelf life.
the potential logistical downsides—reduced availability, increased waste, inventory complexity, and higher costs—likely outweigh the benefits of longer storage for a portion of the platelet supply. Please rewording to “may reducing logistic burden”…..
Readers may also be interested in how 1 or hypothetically 2 weeks of extended storage time solves logistical problems in remote locations?.Please shortly explain
Discussion:
The discussion could generally benefit from some additional commentary regarding earlier studies. A blatant example is the following paragraph, where all the information was shown for the first time (2006-2007) in previously unreferenced studies; “A clear relationship was observed between platelet count and the baseline concentration of soluble factors in the supernatant. At day 21 of storage, RT-stored components exhibited higher baseline concentrations of soluble factors, which coincided with a 50% decrease in the platelet count. Recent studies suggest that ex vivo storage at RT promotes loss of platelet membrane integrity [30], leading to the non-specific release of cytoplasmic contents and soluble factors into the supernatant. In contrast, cold storage leads to less platelet loss, likely due to better preservation of metabolic health, including glucose stores and other metabolites [5, 31, 32]. These findings underscore the role of storage conditions in influencing platelet viability and the release of soluble factors over time”. please rephrase or add pioneer publications
It is reasonable to assume that this paper also becomes more objective by emphasizing that the conclusions are based on a limited number of in vitro parameters.
M&M
If metabolic markers are not reported, how can the authors be certain that the described effects are not originate from an imbalanced metabolism? Please explain or include as limitation
What is the reason why metabolic markers were not tested?
wouldn't ROTEM/TEG have been beneficial for this paper? please comment on that

Author Response
Please see the attachment for responses to Reviewer #1 and #2.

Reviewer 2 Report
Comments and Suggestions for Authors
This article provides some new evidence for the clinical use of a repurposed blood component, the cold-stored PLT. However, in my opinion, some aspects need to be further improved.
Title: In my opinion, the title should say that activation and degranulation on different stimuli are different between cold and RT PLT Intro: The introduction is well written, but the authors should add some elements about new methods to improve cold storage preservation (for example https://pmc.ncbi.nlm.nih.gov/articles/PMC10620573/).
Results Figure 1: b, c, d and LPS should be normalised to the unstimulated state as similarly done for GFs and ILs Figure 2: I would suggest to confirm these data with a functional assay, aggregation or thromboelastography.
Figure 3: y-values should be written as ''Annexin V+ EVs/mL'' and not ''(mL)''.
Table 1: ''Unstimulated'' should be written in the table title Figure S3 should in my opinion be added to the main text given the importance of the PLT count in daily clinical practice.
Tables 2, 3, 4 and 5 should be summarised in a single dot plot showing both significance and delta concentration. In this way the reader can easily visualise the differences between the different conditions, stimuli and between RT and cold PLT. However, I think that these data are very difficult to read.
Discussion
-Authors should state that RT PLT must be used 5-7 days after collection, depending on national legislation.
-When authors say that RT storage (after 21 days) alters membrane integrity by platelet loss, authors should also say that PLT activation does not significantly alter PLT lipid and protein composition as referenced here (https://www.tandfonline.com/doi/full/10.1080/09537104.2023.2281943#d1e815) and thus the difference in terms of PLT ''loss'' between ''specific'' PLT activation (on stimulation) and ''unspecific'' (during storage) needs to be better elucidated.
-The authors found significant differences in RANTES production, how might this correlate with increased inflammation and thus an adverse post-transfusion event? Please discuss.
-In general, I would suggest expanding the discussion in lines 369-380 regarding the GFs and ILs found in the different conditions. The authors state that the proteins selected are markers of inflammation, immune activation and cellular proliferation, but they do not discuss their results in relation to these aspects.
What could be the consequence of increased LPS stimulation? Also in this case an increased pro-inflammatory behaviour of cold stored PLT?
Methods: A representative flow cytometry graph of EV gating should be included in the main or supplementary text.
Author Response

(The authors gave the same response as above.)
